# Timely initiation of postpartum contraceptive utilization in Sebata Hawas district, Ethiopia: A cross-sectional study

**Kamaria Ismael, Tesfaye Getachew Charkos** \*, **Meyrema Abdo**

School of Public Health, Adama Hospital Medical College, Adama, Ethiopia

\* tesfayegch@gmail.com

**Data Availability Statement:** The data used in this paper are in a fully anonymised format at: https://figshare.com/articles/dataset/S1_Data_sav/21541554.

## Abstract

Lack of timely initiating of postpartum contraceptive utilization may lead to mistimed, unintended pregnancies and even become dying as a result of complications related to pregnancy and childbirth. To the best of our knowledge, we have not found enough evidence on the associated factors of timely initiation of postpartum contraceptive utilization in the rural setting of Ethiopia. Therefore, this study aimed to assess the prevalence of timely initiation of postpartum contraceptive utilization and associated factors among women of childbearing age in Sebat Hawas, Oromia, Ethiopia. A community-based cross-sectional study was conducted from March 30 to May 20, 2022. A multistage sampling technique was used to select the participants. Multivariable logistic regression was used to identify associated factors. An adjusted odds ratio with a 95% confidence interval was used to measure the strength of the association. A P-value <0.05 was declared as a statistically significant association. All analysis was performed using SPSS. A total of 804 participants were included in this study. Overall, the prevalence of timely initiation of postpartum contraceptive utilization was 38.6%. In the multivariable models, illiterate women (Adjusted Odd Ratio (AOR): 0.57; 95% CI: 0.35–0.94), with less than 3000 ETB monthly income (AOR: 0.41, 95% CI: 0.22–0.79), counseling on family planning (AOR: 3.75, 95% CI: 1.59–8.83), Menses returned time (AOR: 2.33, 95% CI: 1.15–4.72) and discussion with husband on family planning (AOR: 3.07, 95% CI: 1.61–5.84) were significantly associated with timely initiation of postpartum contraceptive utilization. The findings of this study suggested that the prevalence of timely initiation of postpartum contraceptive utilization was low. Illiterate women, with low monthly income, counseling on family planning, menses returned time, and discussion with their husbands on family planning was the main determinant factors for timely initiation of postpartum contraceptive utilization.

## Background

Globally, Family Planning is recognized as a key life-saving intervention for mothers and their children [1]. WHO recommends that inter-pregnancy intervals should be at least 2 years [2]. Universally around 75% of births are less than 24 months with an interval of fewer than 18

**Funding:** The authors received no specific funding for this work.

**Competing interests:** The authors have declared that no competing interests exist.

**Abbreviations:** AHMC, Adama Hospital Medical College; ANC, Ante-Natal Care; CSA, Central Statistics Agency; EDHS, Ethiopian Demographic and Health Survey; ETB, Ethiopian Birr; FMOH, Federal Ministry of Health; FP, Family Planning; HE, Health Education; HEW, Health Extension Workers; HFs, Health Facilities; HH, House Hold; IRB, Institutional Review Board; IUCD, Intra-Uterine Contraceptive Device; MMR, Maternal Mortality Ratio; MNCH, Maternal Neonatal and Child Health; PNC, Post Natal Care; PPFP, Post-Partum Family Planning; SBA, Skilled Birth Attendant; SDG, Sustainable Development Goal; SRS, Systemic Random Sampling; WHO, World Health Organization.

months between pregnancies associated with an increased risk of low birth weight and small size at birth [3].

Unintended pregnancies particularly among women in developing countries and poor individuals are linked to elevated health problems that resulted in a high number of maternal and neonatal deaths [4, 5]. According to the UN 2015 report an estimated 30 million unplanned births and 40 million abortions, half of them illegal and unsafe, occurred annually in low-and middle-income countries [6]. In sub-Saharan Africa, approximately 53% of women (58 million) who wanted to avoid pregnancy were not using the family planning method [7, 8]. Similarly, another study estimated that 44% of pregnancies were unintended and the prevalence remains high in developing countries, with a quarter of these from Africa [9].

The highest rate of mistimed pregnancy become a big problem, especially in sub-Saharan Africa where approximately half of all pregnancies were reported to have come soon that could have been prevented with increased access to effective utilization of modern postpartum contraceptive methods promptly [10].

Although Post-Partum Family Planning (PPFP) is an increasingly high priority for many countries, uptake and need for PPFP vary. The majority (91%) of postpartum women in low- and middle-income countries (LMIC) report a desire to prevent pregnancy for at least a year following birth [11]. But the result from studies done in low and middle-income countries shows that postpartum modern contraceptive prevalence has been reported to vary from 8% in the first month, 25% in the sixth month to 30% at twelve months after birth [12].

In Ethiopia, of all births in the past five years and current pregnancies, 25% are unintended; from these 17% were mistimed, while 8% are unwanted [13]. A study done in northern Ethiopia shows that nearly half (47%) of all pregnancies occur within a short birth interval of fewer than two years after the recent birth [14]. Studies suggested that the use of contraceptives during the postpartum period helps women to space births by at least 24 months, and this can also help to reduce maternal and child mortalities by 30% and 10%, respectively [15–18]. In addition, contraceptive use during the postpartum period plays a great role in improving the lives of women and their families [19].

Several studies were conducted on the timely initiation of postpartum contraceptive utilization. Some of them are based on small sample size [20] and are unable to conclude the rest of the population. Whereas, in low and middle-income countries like Ethiopia, the majority of the population lived in rural areas with low access to health services. To the best of our knowledge, we have not found enough evidence on the associated factors of timely initiation of postpartum contraceptive utilization in the rural setting of Ethiopia. Therefore, this study aimed to assess the prevalence and associated factors among women of childbearing age in Sebata Hawas District.

## Methods and materials

### Ethics statement

The ethical Committee that approved this study was Institutional Review Board of Adama Hospital Medical College. The reference number was 0914/k373/14. The purpose of the study was explained, and verbal informed consent was taken from each study participants. Information regarding any specific personal identifiers like the name of the participants was not collected, and also the confidentiality of any personal information was also maintained. Each methods were performed based on ethical guidelines and regulations.

### Study design and settings

This cross-sectional study design was conducted from March 30 to May 20, 2022. The study was conducted at Finfinnee special zone, Sebata Hawas district which is located in the central

part of Ethiopia (Fig 1). The district has structured into 2 urban and 34 rural kebeles. In the 2014 census, the total population in Sebata Hawas district was 143,695 people (73,284 males and 70,411 females), of this 31,613 are reproductive age groups. The district has 6 health centers, 36 health posts, and 7 private clinics. All of the facilities are providing maternal and child health services including family planning.

## Study population

Women who gave live birth and were above 42 days postpartum to 12 months have been included in the study. A woman who was died within 12 months after delivery, a woman with a dead birth outcome, seriously ill and can't able to respond to asked questionnaires were excluded from the study.

## Sample size and sampling procedure

The sample size was calculated using a single population proportion formula based on the proportion of timely initiation of postpartum contraceptive utilization in Aroressa district, southern Ethiopia revealing 31.7% of postpartum women [20], 5% level of significance, and a 5% margin of error. Considering a 5% non-response rate and design effects of 1.5, the final calculated sample size was 819.

The stratified multistage sampling technique was used with the strata of the urban and rural kebeles to select 11 rural and 1urban with a simple random sampling technique by using the lottery method. The list of all mothers who gave birth in the previous 12 months was 5504, which was obtained from kebeles extension workers. Then, the total sample size was proportionally allocated to each selected kebeles. A systematic sampling technique was used to select mothers by using their delivery registration list order (S1 Fig).

## Data collection

We collected data through face-to-face interviews using a structured questionnaire. The contents of the variables collected by the questionnaires are maternal age, marital status, educational status of women, place of residence, monthly income, religion, ethnicity, occupational

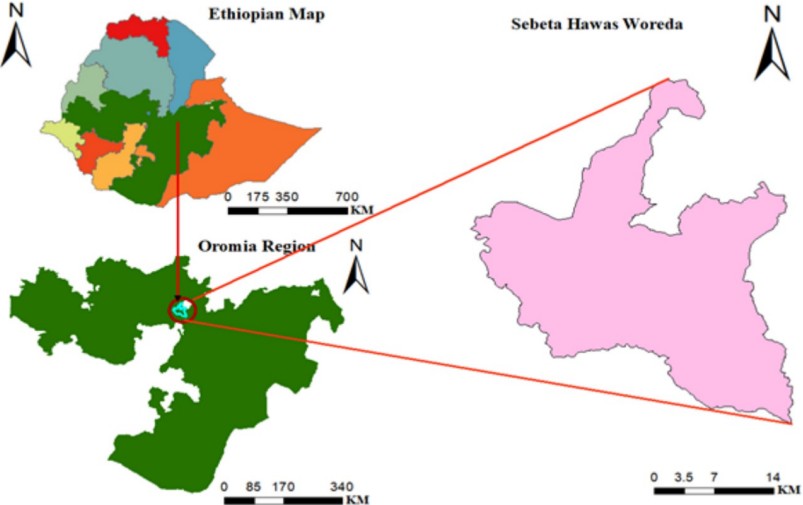

**Fig 1. Study area map of Sebata Hawas districts woreda, Oromia national regional state, Ethiopia in 2022.** The map is drawn by Authors using https://dhsprogram.com/data/dataset/Ethiopia_Interim-DHS_2019.cfm?flag=1.

status of the mother, family size, time menses returned, fertility desire, using contraceptives, the reason for using family planning, type of family planning opted, time of starting modern postpartum contraceptive, ANC follow-up, counseling on family planning, place of delivery, PNC, information, and discussion with their husband.

To assure the quality of the data 5% pre-test of the sample was conducted in the Akaki district near Sebata Hawas, which is in a different population. Training for data collectors on the contents of the questionnaire, the objective of the study, the method of data collection, and interview techniques was done. The collected data were checked for completeness, accuracy, and consistency by supervisors and the principal investigator. The questionnaire was prepared first in English and translated to the local language and was translated back into English.

## Operational definition

Timely initiation of postpartum contraceptive utilization: Starting of modern contraceptive utilization within 6 weeks of the postpartum period.

Hamachisa: is the Oromo cultural norm in which a woman who gives birth stays at the home and doesn't go anywhere until a cultural leader permits or blessed them to do every daily activity. it varies from village to village and the duration of stay is also different, some finish at 2 months of giving birth while others two 6 months.

Discussion with husband: when a woman freely discussed with her husband related to their desire to have children (pregnancy plan), family planning method, and methods of family planning use.

## Data processing and analysis

The questionnaires were checked for completeness, coded, and entered into Epi Info version 3.1 software and then exported to SPSS version 20 for analysis. Descriptive statistics were summarized using frequency and percentage for categorical variables. Bivariate analysis was used primarily to check variables that have an association with the dependent variable individually. Hosmer and Lemeshow's goodness of fit test was checked. Multicollinearity was assessed using variance inflation factors (VIF), where variables with VIF >5 were considered collinear. Variables that were found to have an association with the dependent variable at a P-value of <0.25 were then entered into multivariable logistic regression for controlling the possible effect of confounders and finally the variables which have significant association were identified based on the adjusted odd ratio (AOR), with 95% Confidence Interval (CI) and P-value < 0.05.

## Result

### Socio-demographic characteristics

Eight hundred-four women participated in the study. The median age of the women was 28 years (Inter Quartile Range (IQR): 25–33). Of these, 82.2% were Orthodox followers and 87.2% were Oromo ethnics. Ninety-six percent of these women were married, 65.8% were housewives, 50.7% of them were illiterate, 52.7% had three to six family size, 85% were rural residents and 43.7% of the women had a monthly income <1500 Ethiopian Birr (ETB) (Table 1).

### Characteristics of the women related to maternal health service utilization

In this study, 83.6% of women give live birth. During pregnancy time, 16.9% of the women had two times ANC visits, 24.9% three times, and only 43% of women visited the recommended four times ANC visits. Seventy-nine percent of the women received counseling on family planning during pregnancy, 78.7% of the women delivered at health institution, 81%

**Table 1. Socio-demographic characteristics of the respondents among postpartum mothers in Sebata Hawas district, Oromia central part of Ethiopia 2022.**

| Variable | N | % |
|---|---|---|
| Age (in Year)* | | 28 (25–33) |
| Religion of respondent | | |
| Orthodox | 661 | 82.2 |
| Others | 242 | 17.8 |
| Ethnicity of respondents | | |
| Oromo | 701 | 87.2 |
| Amhara | 44 | 5.5 |
| Others | 59 | 7.3 |
| Marital status | | |
| Married | 775 | 96.4 |
| Others | 29 | 3.6 |
| Educational status | | |
| Illiterate | 408 | 50.7 |
| Literate | 396 | 49.3 |
| Occupation of respondents | | |
| Housewife | 529 | 65.8 |
| Others | 264 | 34.2 |
| Income | | |
| <1500 | 351 | 43.7 |
| 1500–3000 | 254 | 31.6 |
| >3000 | 199 | 24.8 |
| Residence | | |
| Rural | 686 | 85.3 |
| Urban | 118 | 14.7 |
| Family size | | |
| < 3 | 243 | 30.2 |
| 3–6 | 424 | 52.7 |
| >6 | 137 | 17 |

*Median (Interquartile Range (IQR): Q1-Q3)

had postnatal care after the last delivery and only 7.3% of the women had seen their menses less than six weeks. In our sample data, 76.9% of the pregnancy was planned for recent births, and 74.5% of the women utilized contraceptives during study time (Table 2). Of these, 46% use injectables and 43% of the women are using the implant method (Fig 2). In this study, 68.9% of the women discussed with their husbands about Family planning (Table 3).

## Prevalence of timely initiation of postpartum contraceptive utilization

This study revealed that the prevalence of timely initiation of postpartum contraceptive utilization was 38.6% (95%CI: 35.2–42.1). Some of the reasons for not timely initiating postpartum contraceptive utilization are; 32% wait for menses, 25% religious prohibition, 20% cultural reasons like hamachisa, and lack of awareness about the time to initiate (Fig 3).

## Information and awareness about postpartum contraceptive utilization

Almost all 799 (99.4%) respondents heard information about modern postpartum contraception utilization after delivery but from these, those got clear information was 656 (81.6%) and

**Table 2. Characteristics of women related to maternal health service utilization among women who had delivered a baby in the past year in, Sebata Hawas district, Ethiopia 2022.**

| Variable | N | % |
|---|---|---|
| Number of live birth | | |
| 1–4 | 672 | 83.6 |
| > = 5 | 132 | 16.4 |
| ANC follow-up visit | | |
| One visit | 60 | 7.5 |
| Two visit | 136 | 16.9 |
| Three visit | 200 | 24.9 |
| Four visit | 346 | 43 |
| Counseling on family planning | | |
| Yes | 634 | 78.9 |
| No | 170 | 21.1 |
| Place of delivery | | |
| Healthy Facility | 633 | 78.7 |
| Home | 171 | 21.3 |
| PNC follow up | | |
| Yes | 651 | 81 |
| No | 153 | 19 |
| Menses returned time | | |
| < = 6weeks | 59 | 7.3 |
| >6weeks | 410 | 51 |
| Fertility desire | | |
| Yes | 113 | 14.1 |
| No | 691 | 85.9 |
| Pregnancy condition | | |
| Planned | 618 | 76.9 |
| Unplanned | 186 | 23.1 |
| Utilized FP currently | | |
| Yes | 599 | 74.5 |
| No | 205 | 25.5 |

544 (71.4%) of them got information from Health facilities. Out of the total participants, 552 (68.7) of them had awareness about the time to initiate (Fig 4).

## Factors associated with timely initiation of postpartum contraceptive utilization

In multivariable logistic regression analysis; the odds of timely initiating postpartum contraceptive utilization among illiterate women were 43% (AOR: 0.57; 5%CI: 0.35–0.95) less likely to start postpartum contraceptive utilization on time than women who have formal education and similarly, a woman who earned monthly income less than 3000 ETB were 59% (AOR: 0.41; 95%CI: 0.22–0.79) less likely to start postpartum contraceptive utilization on time.

A woman who has received counseling on family planning during pregnancy was 3.7 times (AOR: 3.74; 95% CI: 1.59–8.83) more likely to initiate postpartum contraceptive utilization on time than those who have not received counseling. Women who had returned their menses within the first six weeks after birth were 2.3 times (AOR: 2.33; 95% CI: 1.15–4.72) more likely to initiate postpartum contraceptive utilization on time than those who have not returned their menses within six weeks.

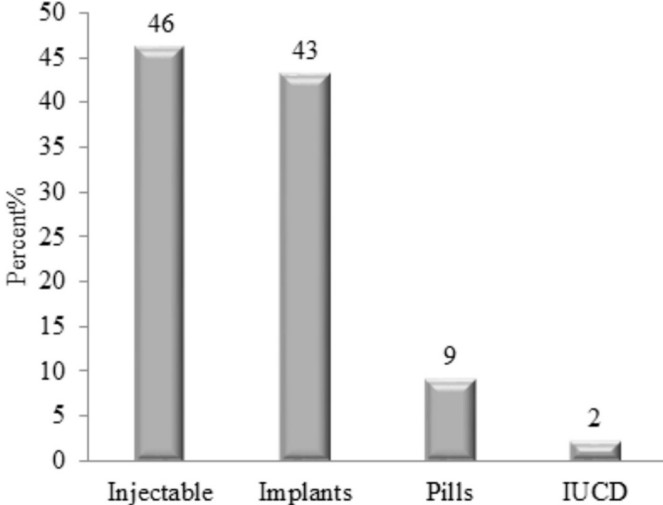

**Fig 2. The distribution of modern contraceptive utilization in Sebata Hawas district, the central part of Oromia, Ethiopia, 2022.**

The odds of timely initiation of postpartum contraceptive utilization among women who had discussed it with their husbands on family planning were 3 times (AOR:3.07, 95%CI: 1.61–5.84) more likely than those who had not discussed it with their husbands (Table 4).

## Discussion

This study was conducted to assess the prevalence of timely initiation of postpartum contraceptive utilization and associated factors among women of childbearing age in Sabata Hawas District, Oromia Ethiopia. Overall, the prevalence of timely initiation of postpartum contraceptive utilization was found to be 38.6%. This finding is consistent with similar studies done in Nigeria [21] and Ethiopia [22], which are 38% and 37.2% respectively. However, this finding is slightly higher than previously done in Ethiopia (31.7%), Malawi (36%), Kenya (35.6%), and India (34.7%) [20, 23–25]. This difference might be due to improvements in health service delivery, the difference in the study period as well as the socio-economic status of the study participants.

The study also identified factors related to the timely initiation of postpartum contraceptive utilization. These include the educational status of the women, monthly income, counseling

**Table 3. Partner communication and discussion towards postpartum contraceptive in Sebata Hawas district, Oromia central part of Ethiopia, 2022.**

| Variable | Category | n | % |
| --- | --- | --- | --- |
| Discussed family planning with Husband | Yes | 554 | 68.9 |
| | No | 250 | 31.1 |
| Husband supported to use family planning method | Yes | 543 | 67.5 |
| | No | 261 | 32.5 |
| Who decides to use Contraceptive | Me my self | 270 | 33.6 |
| | Husband | 80 | 10 |
| | Both together | 415 | 51.6 |
| | Mother in law | 39 | 4.9 |

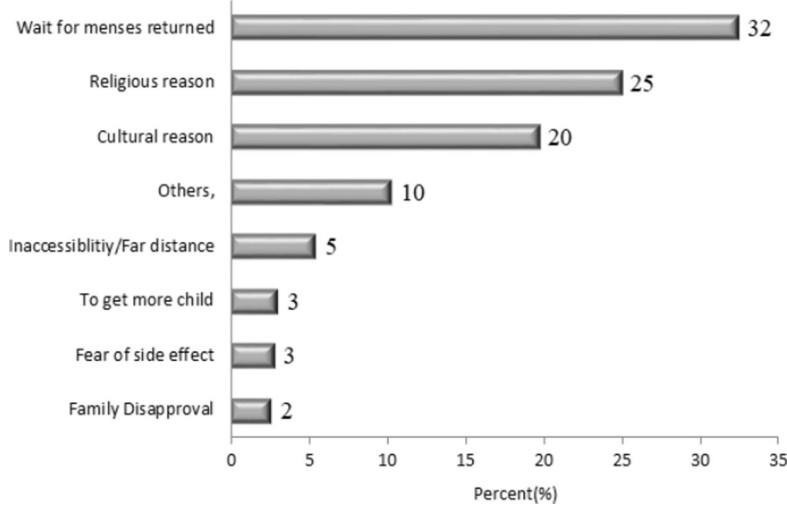

**Fig 3. Reason for not timely initiating postpartum contraceptives utilization in Sebata, Hawas, Oromia, Ethiopia in 2022.**

on family planning during pregnancy, menses returned time, and discussion about family planning with the husband.

This study found that the educational status of postpartum women was significantly associated with the timely initiation of contraceptive utilization. In this study, we found that illiterate woman was 43% (AOR: 0.57; 95%CI: 0.35–0.94) less likely to start postpartum contraceptive on time than literate women. This may have been due to the following reasons. First, as the level of educational attainment increases postpartum women are likely to have a better understanding of the available health facilities and the benefits of fertility regulation [26]. Second, women who have been educated are more likely to visit a health facility and receive counseling or services on family planning and go on to use contraceptives timely, than those who have not been educated [27, 28]. Studies elsewhere have revealed a similar pattern of relationship between educational level and contraceptive use during postpartum [23, 29].

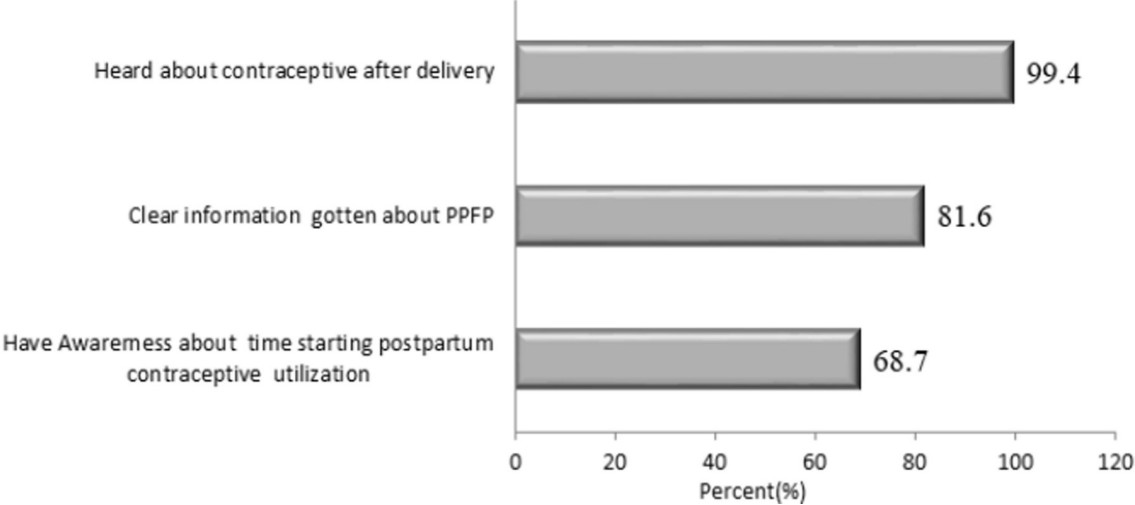

**Fig 4. Information and awareness about PPFP among women who delivered in the past year in Sebata Hawas district, Oromia, Ethiopia 2022.**

**Table 4. Factors associated with timely initiation of postpartum contraceptive utilization among postpartum women in Sabata Hawas district, Oromia central part of Ethiopia, 2022.**

| | Timely initiation of postpartum contraceptive utilization | | AOR (95%CI) | P-value |
|---|---|---|---|---|
| | Yes | No | | |
| Educational status | | | | |
| Illiterate | 98 (24%) | 310 (76%) | 0.569 (0.35,0.94) | 0.026* |
| Literate | 212 (53.5%) | 184 (46.5%) | 1 | |
| Monthly income ETB | | | | |
| <3000 | 182 (30.1.%) | 423 (69.9%) | 0.41 (0.22,0.79) | 0.007** |
| >3000 | 128 (64.3%) | 71 (35.7%) | 1 | |
| Residence | | | | |
| Rural | 244 (35.60%) | 442 (64.4%) | 1.44 (0.62,3.33) | 0.395 |
| Urban | 66 (55.90%) | 52 (44.10%) | 1 | |
| Receive ANC | | | | |
| Yes | 309 (43.1%) | 425 (57.9%) | 1.89 (0.23,14.90) | 0.576 |
| No | 1 (1.4%) | 69 (98.6%) | 1 | |
| Counseling on family planning | | | | |
| Yes | 294 (46.4%) | 340 (53.60%) | 3.75 (1.59, 8.83) | 0.003** |
| No | 16 (9.4%) | 154 (90.60%) | 1 | |
| Place of delivery | | | | |
| Health facility | 298 (47%) | 336 (53%) | 1 | |
| Home | 12 (7.1%) | 158 (92.9%) | 1.11 (0.41, 3.05) | 0.837 |
| Have PNC follow up | | | | |
| Yes | 302 (46.4%) | 349 (53.6%) | 2.75 (0.69, 7.89) | 0.099 |
| No | 8 (5.2%) | 145 (94.80%) | 1 | |
| Menses returned time | | | | |
| < = 6weeks | 38 (64.4%) | 21 (35.6%) | 2.33 (1.15, 4.72) | 0.019 |
| >6weeks | 163 (39.8%) | 247 (60.2%) | 1 | |
| Discussed family planning with husband | | | | |
| Yes | 279 (50.4%) | 275 (49.6%) | 3.07 (1.61, 5.84) | 0.001 |
| No | 31 (12.4%) | 219 (87.6%) | 1 | |

The finding also showed a significant association between monthly income and timely initiation of contraceptives. The odds of timely initiation of postpartum contraceptive use in women who earned a monthly income less than 3000 ETB were 59% (AOR: 0.41; 95%CI: 0.22–0.79) less likely to use than those who earned >3000 ETB. These findings contradict the previous studies [30–33]. This discrepancy might be the study setting, in our case, the study setting is a rural area (85.3%), in which the majority of the women have no formal education and generate less than 3000 birr per month; as a reason due to fair of the cost of service for contraceptive, they might delay.

The finding showed that women whose menses returned earlier after the last delivery were 2.3 times (AOR: 2.33; 95% CI: 1.15–4.72) more likely to initiate postpartum contraceptive utilization on time than those who haven't seen menses after their last delivery. This finding is also consistent with other studies done in Ethiopia [20, 34, 35]. This might be due to the return of menses after delivery alarming women's probability of getting pregnant and this favors the initiation of postpartum contraceptive utilization on time.

Women who received counseling on family planning during pregnancy were 3.7 times (AOR: 3.75; 95% CI:1.59–8.83) more likely to initiate postpartum contraceptive utilization on time than those not received counseling. This may be because women who received

postpartum contraceptive counseling during pregnancy or delivery might be highly motivated to use modern contraceptive methods. This relationship is supported by a study done in Ethiopia [22, 36].

The woman who discussed family planning with their husband or partners was 3 times (AOR: 3.0; 95%CI: 1.61–5.84) more likely to initiate postpartum contraceptive utilization timely than those who did not discuss family planning with their husband. The possible explanation for this might be that women can get more information and support to utilize maternal health services through discussing with their husbands/partners, so this might increase their intention to access contraceptive methods in an efficient and timely manner after delivery. These are findings in line with the previous studies [20, 37].

This study has some limitations. First: the study was a cross-sectional study design, in which the causal relationship between dependent and independent variables could not be established. Second: recall bias might be introduced, as a reason to some extent the robustness of the results may be affected. Lastly: this study did not address all health system-related factors that affect the timely initiation of postpartum contraceptive utilization, because we focus mainly on the quantitative approach.

In conclusion, the prevalence of timely initiation of postpartum contraceptive utilization in the Sebeta Hawas district was low. Although we investigated that illiterate women, monthly income less than 3000 EBR, counseling on family planning during pregnancy, menses returned time, and discussion on the family planning with husband was significantly associated with timely initiation of postpartum contraceptive utilization among women who are the residence of Sebata Hawas district. Therefore, healthcare providers should strengthen the integration of family planning services with maternal and child health services, provide health information about the timely utilization of contraceptives and improve postnatal care follow-up after giving birth.

## Supporting information

**S1 Fig. Scheme presentation of sampling procedure in the Sebata Hawas, Oromia, Ethiopia.**
(TIF)

## Acknowledgments

The authors like to acknowledge the study participants, data collectors, colleagues, and the Adama Hospital Medical College.

## Author Contributions

**Conceptualization:** Tesfaye Getachew Charkos.

**Data curation:** Kamaria Ismael.

**Formal analysis:** Kamaria Ismael, Tesfaye Getachew Charkos, Meyrema Abdo.

**Investigation:** Kamaria Ismael, Tesfaye Getachew Charkos, Meyrema Abdo.

**Methodology:** Kamaria Ismael, Tesfaye Getachew Charkos, Meyrema Abdo.

**Project administration:** Tesfaye Getachew Charkos, Meyrema Abdo.

**Resources:** Kamaria Ismael.

**Software:** Kamaria Ismael, Tesfaye Getachew Charkos.

**Supervision:** Tesfaye Getachew Charkos, Meyrema Abdo.

**Validation:** Tesfaye Getachew Charkos.

**Visualization:** Kamaria Ismael, Tesfaye Getachew Charkos.

**Writing – original draft:** Kamaria Ismael, Tesfaye Getachew Charkos, Meyrema Abdo.

**Writing – review & editing:** Kamaria Ismael, Tesfaye Getachew Charkos, Meyrema Abdo.

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
