## [Decision Letter · Decision Letter 0]

2 Oct 2022

PGPH-D-22-01272

Why timely initiation of postpartum contraceptive utilization in Ethiopia is low? a cross-sectional study

Dear Dr. Charkos,

Thank you for submitting your manuscript to PLOS Global Public Health. After careful consideration, we feel that it has merit but does not fully meet PLOS Global Public Health’s publication criteria as it currently stands. Therefore, we invite you to submit a revised version of the manuscript that addresses the points raised during the review process.

We look forward to receiving your revised manuscript.

Kind regards,

Kimiyo Kikuchi

Academic Editor

Journal Requirements:

1. We have amended your Competing Interest statement to comply with journal style. We kindly ask that you double check the statement and let us know if anything is incorrect. 

2. We are unable to open your Supporting Information file Supporting information.rar. Please kindly revise as necessary and re-upload.

3. In the online submission form, you indicated that "The data used to support the findings of this study are available from the corresponding author upon reasonable request.We have also attached as per requested." All PLOS journals now require all data underlying the findings described in their manuscript to be freely available to other researchers, either 1. In a public repository, 2. Within the manuscript itself, or 3. Uploaded as supplementary information.

4. Some material included in your submission may be copyrighted. According to PLOS’s copyright policy, authors who use figures or other material (e.g., graphics, clipart, maps) from another author or copyright holder must demonstrate or obtain permission to publish this material under the Creative Commons Attribution 4.0 International (CC BY 4.0) License used by PLOS journals. Please closely review the details of PLOS’s copyright requirements here: PLOS Licenses and Copyright. If you need to request permissions from a copyright holder, you may use PLOS's Copyright Content Permission form.

Potential Copyright Issues:

Figure 1: please (a) provide a direct link to the base layer of the map (i.e., the country or region border shape) and ensure this is also included in the figure legend; and (b) provide a link to the terms of use / license information for the base layer image or shapefile. We cannot publish proprietary or copyrighted maps (e.g. Google Maps, Mapquest) and the terms of use for your map base layer must be compatible with our CC-BY 4.0 license. 

Additional Editor Comments (if provided):

Here are additional comments from the editor:

Complete English editing is required. 

Abstract: Wouldn't it be more appropriate for the "OR" to be "AOR"?

L47, L120 Please spell out any abbreviations appearing in the text for the first time.

L72 Figure 1 does not show the study period.

L77-L78 The sentence "All woman who…" should be rewritten.

L104-109 Explanations are needed on how "time menses returned" and "time of starting modern postpartum contraceptive" were measured. "Discussion with husband" on what?

Method: Did the authors conduct a face-to-face interview using a questionnaire or conduct a self-administered questionnaire?

L131-134、139、142、144、148、149　 The positions of the numbers are inappropriate. L147－149　Please rewrite the sentence.

L157　Is the parentheses of 32％ necessary?

L158  Need explanation of "hamachisa".

L156-159 Incorrect sentence

L162-164　Incorrect sentence

L178　Incorrect parentheses

L191　Since this is a cross-sectional study, "affect" is not an appropriate term. Better to use  "related to" instead.

The authors wrote the references in L204, but please write the relevant references after the first reason (L199-201) and the second reason(L201-203), respectively.

L208―210　Incorrect sentences.

L208-210 From the odds results, we understand that women with lower incomes were less likely to use contraception. However, the author's discussion seems to argue the opposite result.

L217-219　Incorrect sentence. Which is the subject?

L223-225　Incorrect sentence

L226　 What exactly does "more information" mean in the context of a contraceptive?

Reviewers' comments:

Reviewer's Responses to Questions

**Comments to the Author**

1. Does this manuscript meet PLOS Global Public Health’s publication criteria? Is the manuscript technically sound, and do the data support the conclusions? The manuscript must describe methodologically and ethically rigorous research with conclusions that are appropriately drawn based on the data presented.

Reviewer #1: Partly

Reviewer #2: Partly

2. Has the statistical analysis been performed appropriately and rigorously?

Reviewer #1: I don't know

Reviewer #2: Yes

3. Have the authors made all data underlying the findings in their manuscript fully available (please refer to the Data Availability Statement at the start of the manuscript PDF file)?

Reviewer #1: Yes

Reviewer #2: Yes

4. Is the manuscript presented in an intelligible fashion and written in standard English?

Reviewer #1: No

Reviewer #2: Yes

5. Review Comments to the Author

Reviewer #1: Manuscript Number: PGPH-D-22-01272

Title: Why timely initiation of postpartum contraceptive utilization in Ethiopia is low? a cross-sectional study

The topic peaked out by the authors is vital for developing countries like Ethiopia where a major proportion of the population resides in a rural setting with limited health services, uneducated mothers, etc.

Thus, dealing with the actual status of timely initiation of postpartum contraceptive utilization and associated factors is important to make the tailored intervention.

However, the manuscript needs major revision in its content.

General

The typo of the manuscript needs reworks – line space, comma use, inconsistency in using upper and lower cases (both in paragraphs and headings/subheadings), and repetition of ideas.

Methods and Materials

1. Line 68 – 70, the sentence structure needs rephrasing (like ‘Based on… of which 31613 are reproductive…’

2. Under the subheading “study population description” – It is better to merge two sentences and rephrase it.

3. Sample size and sampling procedure - parts of the first paragraph is repeated in the second paragraph

Results

- In general, the results part needs deep revision in terms of narration approach and grammar. To point out the specific areas some highlights are made on the mother document for the authors' perusal.

- Subheading ‘Maternal health Service utilization related’ looks incomplete

- Better to merge subheading ‘Partner communication and discussion’ to ‘Maternal health Service utilization related’ – consider rephrasing the sub-heading

Discussion

- Line 196 – 198 & 206 – 208: seems a repetition of the results.

- The discussion part needs a refinement

- The authors have mentioned one of the study's strengths is being a study of a large scale. Since the study covered one district only, what makes it large-scale?

- Line 234 & 235: Better to rephrase like “……between the independent and dependent variable could not be established.”

Conclusion

- No need of repeating figures which were already given in the result part.

Reference

- Need rework to make it up to the standard of a scientific paper

Questions

Educational status of the women, monthly income, counseling about family planning during pregnancy or delivery, menses returned time, and discussion about family planning with husband reported to improve timely initiation of PPFP. What is new here? Is it not a natural phenomenon? Do the authors encounter any findings indicating that these variables have a negative influence? What message the authors are going to convey from this study?

Reviewer #2: Comments to the Author

This manuscript described “Why timely initiation of postpartum contraceptive utilization in Ethiopia is low? a cross-sectional study” which is an important issue to assess the timely initiation of PPFP utilization particularly in the study area. Despite the interesting scope of the research, the manuscript needs revision. Authors expected to explain and clarify the comments written below.

Title: The title is appropriate, but needs to be clearly specify the study settings.

- What is ‘‘Timely initiation of contraceptive utilization’’ Is it Immediate PPFP? Contraceptive utilization within 6 weeks of postpartum period? – if yes why did you intended to conduct within 6 weeks than Immediate?

Abstract

-- Background: Please add a clear rationale for conducting this study. There is a lot studied done on PPFP.

- Result: …..Timely initiation of postpartum utilization…rewrite as Timely initiation of postpartum contraceptive utilization

- Conclusion and recommendation: write your recommendation based the findings of the study. Do you think 38.6% of PPFP utilization is considered as low?

Introduction

1. L60-62: What makes your study differ from others? Please write clear justification

Methods

1. Put the study design

2. Why you include women who gave birth in the last 12 months before data collection and exclude women with dead birth outcome and were less than 6 weeks of postpartum period?

3. Is 5504 women gave birth at Sabata Hawas District or at 12 selected areas per 12 months?

4. Where do you conduct the pretest?

5. Please clearly write the operational definition of the study…Timely initiation of postpartum contraceptive utilization…..

6. How did you use …using contraceptives, reason for using family planning, type of family planning opted, time of starting modern postpartum contraceptive as independent variables?

7. L141-142: Received counseling about FP… When? A. during ANC? B. during labour and delivery? C. during PNC D. during all cascade of pregnancy (ANC, labour and delivery, and PNC)?

Discussion

1. L186-190: the discussion needs logical justification

2. L217: Counselling during pregnancy/delivery VS L141:- Counselling during pregnancy Vs Table2: Counselling on FP ….lacks consistency

Conclusion:

- Add - Conclusion and recommendation section

- Needs to be more punchy and to the point/ findings.

? Table 2: Do you think number of live birth or number of having alive children determine utilization?

6. PLOS authors have the option to publish the peer review history of their article (what does this mean?). If published, this will include your full peer review and any attached files.

**Do you want your identity to be public for this peer review?** For information about this choice, including consent withdrawal, please see our Privacy Policy.

Reviewer #1: No

Reviewer #2: **Yes: **Mulualem Silesh

---

## [Decision Letter · Decision Letter 1]

2 Dec 2022

PGPH-D-22-01272R1

Timely initiation of postpartum contraceptive utilization in Sebata Hawas district, Ethiopia: a cross-sectional study

Dear Dr. Charkos,

Thank you for submitting your manuscript to PLOS Global Public Health. After careful consideration, we feel that it has merit but does not fully meet PLOS Global Public Health’s publication criteria as it currently stands. Therefore, we invite you to submit a revised version of the manuscript that addresses the points raised during the review process.

We look forward to receiving your revised manuscript.

Kind regards,

Kimiyo Kikuchi

Academic Editor

Journal Requirements:

Additional Editor Comments (if provided):

It is much better but does not fully answer some of the reviewers' comments. For example, the exclusion of dead women must be adequately stated in the paper, as this could cause bias in the study results.

Generally, the study design is written at the beginning of the methods. Therefore, I recommend that the following modifications be made.

- Change the subheading "Study area and period" to "Study design and settings".

- Start the sentence of line 69 with the study design, such as "This cross-sectional study was conducted…" or "This is a cross-sectional study which conducted…".

The data collection methods should also be clearly stated in the methods section. Therefore, I suggest the authors make a subheading "Data collection" in the methods section and start with a statement such as "We collected data by face-to-face interview using a structured questionnaire." Also, I recommend moving descriptions of the contents of the variables collected by the questionnaires, translations of the questionnaire, and data collector training in the section of the "data collection" subheadings.

Reviewers' comments:

Reviewer's Responses to Questions

**Comments to the Author**

1. If the authors have adequately addressed your comments raised in a previous round of review and you feel that this manuscript is now acceptable for publication, you may indicate that here to bypass the “Comments to the Author” section, enter your conflict of interest statement in the “Confidential to Editor” section, and submit your "Accept" recommendation.

Reviewer #2: (No Response)

2. Does this manuscript meet PLOS Global Public Health’s publication criteria? Is the manuscript technically sound, and do the data support the conclusions? The manuscript must describe methodologically and ethically rigorous research with conclusions that are appropriately drawn based on the data presented.

Reviewer #2: Yes

3. Has the statistical analysis been performed appropriately and rigorously?

Reviewer #2: Yes

4. Have the authors made all data underlying the findings in their manuscript fully available (please refer to the Data Availability Statement at the start of the manuscript PDF file)?

Reviewer #2: Yes

5. Is the manuscript presented in an intelligible fashion and written in standard English?

Reviewer #2: Yes

6. Review Comments to the Author

Reviewer #2: Thank you for your response. The topic is very important as input for policy makers, program managers and others stack holders particularly for study settings. However, still the manuscript needs revision and all comments are not addressed.

1. Why you exclude women with dead birth outcome? Do you think FP is not recommended to initiate early for those women who had died birth outcome?

2. Sampling procedure is not clear. Please put the schematic presentation of sampling technique 3. Why did you intended to conduct within 6 weeks than immediate postpartum period? I think it’s better to rewrite the title as postpartum contraceptive Utilization …..

4. How you consider contraceptives, reason for using family planning, type of family planning opted, time of starting modern postpartum contraceptive as independent variables?

5. Received counseling is still lacks consistency throughout the manuscript. How you consider/categorized as woman had received counseling on FP or not?

7. PLOS authors have the option to publish the peer review history of their article (what does this mean?). If published, this will include your full peer review and any attached files.

**Do you want your identity to be public for this peer review?** For information about this choice, including consent withdrawal, please see our Privacy Policy.

Reviewer #2: **Yes: **Mulualem Silesh

---

## [Editor Report · Decision Letter 2]

27 Dec 2022

Timely initiation of postpartum contraceptive utilization in Sebata Hawas district, Ethiopia: a cross-sectional study

PGPH-D-22-01272R2

Dear Dr. Charkos,

We are pleased to inform you that your manuscript 'Timely initiation of postpartum contraceptive utilization in Sebata Hawas district, Ethiopia: a cross-sectional study' has been provisionally accepted for publication in PLOS Global Public Health.

Best regards,

Kimiyo Kikuchi

Academic Editor